# Testicular Neoplasms: Primary Tumour Size Is Closely Interrelated with Histology, Clinical Staging, and Tumour Marker Expression Rates—A Comprehensive Statistical Analysis

**DOI:** 10.3390/cancers14215447

**Published:** 2022-11-05

**Authors:** Klaus-Peter Dieckmann, Hendrik Isbarn, Francesca Grobelny, Cansu Dumlupinar, Julia Utschig, Christian Wülfing, Uwe Pichlmeier, Gazanfer Belge

**Affiliations:** 1Department of Urology, Asklepios Klinik Altona, D-22763 Hamburg, Germany; 2Martini Klinik, Universitätsklinikum Eppendorf, Martinistrasse 52, D-20251 Hamburg, Germany; 3Tumour Genetics Department, Faculty of Biology and Chemistry, Tumour Genetics, University of Bremen, Leobener Strasse 2/FVG, 28359 Bremen, Germany; 4Institute of Medical Biometry und Epidemiology, Universitätsklinikum Eppendorf, Martinistrasse 52, D-20251 Hamburg, Germany

**Keywords:** testicular tumour, seminoma, tumour marker, microRNA, tumour size, germ cell tumour, alpha-fetoprotein, human chorionic gonadotropin

## Abstract

**Simple Summary:**

The size of the tumour is of clinical relevance in testicular tumours. Our statistical analysis of 641 cases with testicular tumours revealed that 50% of cases with tumours smaller than 1 cm were of benign histology. The frequencies of both metastatic spread and elevations of serum tumour markers increase with growing tumour size. The novel tumour marker microRNA-371a-3p (M371) outperforms the classical markers and is measurable in 44% of germ cell tumours with tumour sizes below 1 cm.

**Abstract:**

The role of primary tumour size (TS) in the clinical course of testicular tumours is incompletely understood. We retrospectively evaluated 641 consecutive patients with testicular neoplasms with regard to TS, histology, clinical stage (CS), serum tumour marker (STM) expression and patient age using descriptive statistical methods. TS ≤ 10 mm was encountered in 13.6% of cases. Median TS of 10 mm, 30 mm, 35 mm, and 53 mm were found in benign tumours, seminomas, nonseminomas, and other malignant tumours, respectively. In cases with TS ≤ 10 mm, 50.6% had benign tumours. Upon receiver operating characteristics analysis, TS of > 16 mm revealed 81.5% sensitivity and 81.0% specificity for detecting malignancy. In subcentimeter germ cell tumours (GCTs), 97.7% of cases had CS1, and CS1 frequency dropped with increasing TS. Expression rates of all STMs significantly increased with TS. MicroRNA-371a-3p (M371) serum levels had higher expression rates than classical STMs, with a rate of 44.1% in subcentimeter GCTs. In all, TS is a biologically relevant factor owing to its significant associations with CS, STM expression rates and histology. Importantly, 50% of subcentimeter testicular neoplasms are of benign nature, and M371 outperforms the classical markers even in subcentimeter tumours.

## 1. Introduction

The clinical status of testicular tumours is characterized by the parameters primary tumour size, patient age, histology of the primary, serum tumour marker elevations and the extent and distribution of metastases (i.e., clinical staging). These features represent simple clinical factors, and the clinical usefulness of each factor has stood the test of time [1,2,3,4,5] despite ongoing incorporation of molecular genetic factors in other malignancies [6]. However, only modest information is available regarding the biological interrelationships between the factors. In particular, the clinical and biological role of primary tumour size is still incompletely understood. The size of a testicular new growth is considered to have a bearing on histology found upon surgery [7]. This knowledge is of high clinical relevance, because in light of the continuously growing use of scrotal ultrasound in everyday urological practice and ongoing technical refinements of this technology [8,9,10], the incidence of small testicular neoplasms has been increasing in recent years. Tumour size also has a bearing on metastatic spread, at least in the subgroup of seminoma [11]. Likewise, primary tumour size is associated with the frequency of serum tumour marker elevations (expression rates) and the extent of elevations of classical serum tumour markers alpha fetoprotein (AFP), beta human chorionic gonadotropin (bHCG) and lactate dehydrogenase (LDH) [12]. Additionally, there is growing evidence for an association of the novel serum tumour marker microRNA-371a-3p (M371) with tumour bulk in general and tumour size in particular [13,14]. Finally, there is some suggestion of an association of patient age and tumour size, with a trend towards larger tumours in the elderly [15].

Obviously, there are manifold interrelationships between tumour size and other clinical characteristics of testicular neoplasms. A number of studies had analysed single issues regarding the clinical role of tumour size [12,16,17,18], but so far, there is no comprehensive systematic analysis regarding the clinical significance of this factor.

Therefore, we analysed tumour size in a large patient sample and looked for associations with histology of the primary, clinical staging, serum tumour marker expression rates, and with age. In particular, we sought to assess the following four assumptions, most of which are already supported by some pieces of evidence; however, the levels of evidence have not reached high levels so far: *(Assumption #1*) The size of a testicular new growth is associated with histology in the way that a much higher proportion of benign tumours are found among primary testicular neoplasms sized < 1 cm than among the larger ones. (*Assumption #2*) Primary tumour size has an impact on clinical staging in germ cell tumours (GCTs), with fewer advanced clinical stages in small tumours and more advanced stages in larger tumours. (*Assumption #3*) Tumour size is associated with the expression rates of the classical serum tumour markers in GCTs as well as with the novel marker M371. Lower frequencies of marker elevations are expected in small tumour sizes, and higher expressions in larger tumours. The new tumour marker M371 is much more frequently expressed than the classical markers bHCG and AFP. *(Assumption #4)* An interrelationship between patient age and primary tumour size is hypothesized in a way that older patients will have larger tumours.

## 2. Materials and Methods

### 2.1. Patient Recruitment, Data Procurement

All consecutive patients, aged 17–98 years, undergoing surgery for a newly diagnosed testicular mass in two Hamburg-based departments of general urology (Albertinen-Krankenhaus and Asklepios Klinik Altona) during 2012–2021, were included in the present retrospective study. Patients submitted to orchiectomy after previous chemotherapy were excluded. The following parameters were abstracted from archival case files: size of the testicular mass (mm) as recorded in the pathology reports, patient’s age (years), histology of the surgical specimen categorized as seminoma (SE), nonseminoma (NS), benign tumour (BT), malignant tumour other than GCT (OM), clinical stage (CS) (only in GCTs); and preoperative expression of serum tumour markers bHCG, AFP, LDH, and M371. About one third of the patients had been included in previous investigations on various other issues [12,14,19].

In case of multiple separate tumours in one testicle, the diameters of each tumour were added up to the final tumour size. Simultaneous bilateral tumours were excluded from this analysis. Histological diagnoses were retrieved from pathology reports without central pathology review. Clinical stages were recorded as CS1; CS2a,b; CS2c; and CS3 according to current guide-lines [20]. Serum tumour markers bHCG, AFP, and LDH were measured in hospital laboratories according to institutional guidelines. As during the observation period the commercially available test kits were changed for economic reasons with consecutive changes in upper limits of norm (ULN), we only recorded elevations of serum levels above ULN (yes/no). M371 serum levels were measured as detailed previously [14]. Briefly, RNA was isolated from aspirated serum, followed by reverse transcription to cDNA for both miR-371a-3p and the endogenous control miR-30b-5p. The quantitative polymerase chain reaction was performed after preamplification. M371 serum levels were originally provided as relative quantity (RQ) values. However, to be methodologically consistent with the analysis of classical markers, only dichotomized results were noted (elevated yes/no) by defining RQ = 5 as ULN.

Ethical approval was provided by Ethikkommission der Ärztekammer Hamburg on 2 March 2020 (PV7288). As solely anonymized patients’ data were evaluated in the present study, the need for informed consent of patients was waived by the ethical committee. All study activities were conducted according to the Declaration of Helsinki of the World Medical Association as amended by the 64th General Assembly, October 2013. 

### 2.2. Statistical Analysis

All patient data were originally filed in a commercially available data base (MS Excel, version 2017) and subjected to data validation prior to statistical analysis. For descriptive analysis of nominal variables, absolute frequencies, percentages and 95% confidence intervals (CIs) were presented. 

Continuous variables were descriptively analysed by calculating measures of location and dispersion, such as median, first quartile (Q1), third quartile (Q3), interquartile range (IQR), minimum, maximum, arithmetic mean and standard deviation. For graphical presentation, box and whisker diagrams were created; the respective boxes were defined by Q1, median and Q3, and the whiskers were defined by the largest and lowest observed values that fell within the 1.5 times IQR measured from Q3 and Q1, respectively. 

To test for any differences in the distribution of tumour sizes between more than two subgroups of patients defined by age and histology, Kruskal–Wallis tests were applied; these were replaced by Wilcoxon two-sample tests in case of two categories. Chi-squared tests were used to compare contingency tables of nominal variables. To assess whether malignancy rates of testicular neoplasms or tumour marker expression rates increased with tumour size, Cochran–Armitage trend tests were applied. The Jonckheere–Terpstra test served to assess whether tumour sizes increased with clinical stage. 

To assess the ability of the tumour size to predict malignant histology, a receiver operating curve (ROC) was created. The area under the curve (AUC), together with its 95% Wald confidence interval, was calculated to quantify the goodness of prediction. Youden’s index, defined as the maximum vertical distance between the ROC curve and the diagonal line, was used to identify the optimal cutoff of the tumour size for malignancy prediction. Additionally, the predicted probability curve for malignant histology with bounding 95% confidence intervals was calculated using a linear logistic regression model. 

Due to the high completion rate of patient data sets, all patients were eligible for most of the analyses listed above. Only serum tumour marker expressions (especially for M371 marker) were not available in all patients. Thus, respective statistical analyses were performed with varying sample sizes according to the available entries. 

*p*-values of less than 5% were considered statistically significant in this paper.

Statistical analysis was performed with SAS software package version 9.4 (SAS Institute, Cary, NC, USA) on the Windows platform. 

## 3. Results

### 3.1. General Results

A total of 641 patients with a median age of 38 years were included in the present analysis. The frequencies of the four histologic subgroups with corresponding median ages are given in Table 1. Histologically, benign tumours (BT) comprised gonadal stromal tumours for the most part and benign epidermoid cysts and other rare tumours to a lesser degree. Other, malignant tumours (OM) mainly comprised diffuse large B-cell lymphomas with few other forms of malignant testicular lymphoma. Age, tumour size, histology, and clinical staging were available in all patients. Marker elevations regarding AFP and bHCG were available in 640 patients, regarding LDH in 633 cases, and with respect to M371 in 451 patients. The median tumour size of the entire population (*n* = 641) was 30 mm (IQR 15–45 mm). Tumour sizes of ≤10 mm were found in 13.6% of patients, while 45.2% of patients presented with tumour sizes > 30 mm. 

### 3.2. Assumption # 1 (Association of Tumour Size with Histology)

The median tumour sizes found in germ cell tumours and in the four histologic subgroups are delineated in Table 2 and in Figure 1. The largest median tumour size, of 53 mm, was observed in the other malignant tumours (OM). The largest tumour size over-all was 18.9 cm in a patient with nonseminoma (so-called giant tumour). Overall, median tumour sizes were significantly different among the four histologic groups (Kruskal–Wallis test *p* < 0.0001). 

Results of the comparisons of median tumour sizes between particular histological subgroups are listed in Table 3.

Tumour sizes of seminomas and nonseminomas were not significantly different from each other. However, median GCT tumour size (SE + NS) was significantly larger than that of benign tumours (BT), while OM were significantly larger than GCTs (SE + NS) and BT. A typical example of a small benign tumour is given in Figure 2. 

The frequencies of the four histologic subgroups in tumour size categories ≤ 10 mm and >10 mm, respectively, are shown in Table 4 and Figure 3. The distribution of histologic subgroups was significantly different between the two size categories. 

In the subcentimeter category, benign tumours represented more than half of the cases. In the larger tumour size category, seminomas predominated with 59.6%, while benign tumours comprised only 6.3%.

A more granular analysis with histologic subgroup frequencies in tumour size categories ≤ 10 mm; 11–20 mm; 21–30 mm; and >30 mm is provided in Table 5. 

Overall, the distribution of histologic subgroup frequencies was significantly different among the tumour size categories (*p* < 0.0001, chi square test). If benign tumours (BT) were compared with all malignant tumours (GCT + OM), it became clear that in subcentimeter testicular neoplasms, more than half of all cases represented benign tumours, while decreasing relative proportions of this subtype were observed in categories with increasing tumour sizes (Table 6).

An inverse distribution of frequencies was found among the malignant tumours. The trend towards lower proportions of BT and higher frequencies of malignant tumours (GCT + OM) in increasing tumour size categories was highly significant (*p* < 0.0001, Cochran–Armitage trend test). The ROC analysis in Figure 4 exhibits the ability to diagnose a testicular neoplasm as a benign tumour by means of tumour size. Using a tumour size of 16 mm as threshold between benign and malignant tumours, the sensitivity and specificity of this factor were 81.5% and 81.0%, respectively (highest Youden index: 0.62507), with an area under the curve (AUC) of 0.8912 (95% Wald CI, 0.8569–0.9256). 

The ROC curve with an AUC of 0.8912 showed that tumour size is in fact a useful tool for assessing the biologic behaviour of small testicular neoplasms. Using a tumour size of 16 mm as threshold between benign and malignant tumours, the sensitivity and specificity of this factor were 81.5% and 81.0%, respectively (Youden index: 0.62507).

Figure 5 presents a logistic regression analysis for predicting malignancy with tumour size. In this model, all tumour sizes ≥39 mm indicated malignancy, while tumours sized ≤8 mm involved a 50% chance of being malignant. 

### 3.3. Assumption #2 (Association of Tumour Size and Clinical Staging in GCTs)

Table 7 and Table 8 list the frequencies of clinical stages in four tumour size categories of germ cell tumours. Notably, in the smallest tumour size category (≤10 mm), almost 98% of cases had clinical stage 1. Overall, CS frequencies were significantly different among the tumour size categories (*p* < 0.0001, Kruskal–Wallis test). Table 8 shows that among CS1 cases, there was a significant trend towards decreasing frequencies of cases with increasing tumour size, and an inverse trend was seen in stages with metastases (*p* < 0.0001; Cochran–Armitage trend test). 

The median tumour sizes with IQR and ranges in the four clinical stages are provided in Table 9 and Figure 6. The median tumour size increased with clinical stage, and overall, median tumour sizes were significantly different among the four CS (*p* < 0.0001, Kruskal–Wallis test).

### 3.4. Assumption #3 (Tumour Size Is Associated with Frequencies of Tumour Marker Expression in GCTs)

Table 10 outlines the frequencies of elevations of the serum tumour markers bHCG, AFP, LDH, AFP and/or bHCG, and M371 found in the entire population of patients stratified for tumours sized ≤10 mm and >10 mm, respectively. All of the markers were significantly more frequently expressed in larger than in smaller tumour sizes (all comparisons *p* < 0.0001, chi square test).

M371 had the highest expression rate of all tumour markers in both size categories. Somewhat higher expression rates were observed if only germ cell tumours were evaluated (Table 11), but again, all markers had significantly higher expression rates in patients with tumour sizes >10 mm than in those with smaller sizes. The highest expression rate overall was 90.15%, found for M371 in GCTs sized >10 mm. 

Table 12 shows the results for the tumour marker expression levels in the subgroup of GCT patients with CS1 stratified by tumour sizes ≤10 mm and >10 mm, respectively. Again, marker expression levels were higher in the category with larger tumour sizes. However, regarding LDH, the difference was no more significant (*p* = 0.1039, chi-squared test). Notably, 44% of CS1 patients with tumours ≤10 mm showed elevations of M371 levels, while each of the other markers were expressed in less than 10%.

Table 13 delineates the expression rates of all tumour markers stratified by five tumour size categories in the entire study population and in the histologic subgroups of seminoma and nonseminoma (Figure 7). In all subpopulations, the expression rates of all tumour markers increased with increasing tumour sizes, except for AFP in the seminoma subgroup (Cochran–Armitage trend test, all except seminoma *p* < 0.0001; seminoma *p* = 0.9133). M371 revealed the highest expression rates in all size categories. 

### 3.5. Assumption #4 (Association of Patient Age with Tumour Size)

Table 14 outlines the median tumour sizes in four age categories of the entire study population of patients with testicular neoplasms. Overall, the median tumour sizes were significantly different among the age categories, with the largest size in the age category >50 years (Kruskal–Wallis test *p* = 0.0117).

However, there was no clear trend towards larger tumours in older patients or vice versa. Table 15 lists the median tumour sizes in the four age categories in patients with germ cell tumours. Here again, the largest median tumour size was observed in the oldest age category. Overall, the difference between the tumour sizes among the age categories was significant (*p* = 0.0161), but the difference between the youngest and the oldest age categories was only 1.5 years, and the inter quartile ranges widely overlapped. Thus, no clear association between age and tumour size could be documented. 

## 4. Discussion

The present study thoroughly analysed the clinical influence of tumour size on presenting features of testicular neoplasms. One main result was that subcentimeter tumours comprised around 13.6% of all testicular new growths, which represented a frequency that had not been clearly specified before. Other central results were the significant associations of primary tumour size with the three factors histology of the primary, clinical staging, and serum tumour marker expression. 

### 4.1. Tumour Size—General Considerations

The median sizes of seminomas and nonseminomas in the present study were 30 mm and 35 mm, respectively, with no significant difference between the two subtypes. This result is consistent with sizes of 35 mm found in both seminoma and nonseminoma in a Swiss study [21]. Similar results were reported in studies from Mannheim University with 37 mm (SE) and 38 mm (NS) [17,22], while a Cologne-based study reported a comparatively small mean size of 15 cm^3^ for GCTs, which would correspond to a mean diameter of only 25 mm [23]. From Texas, a slightly higher mean diameter of 41 mm was reported for Americans of Caucasian descent [24]. However, as also shown in the present analysis, mean size may be somewhat larger than the median value. For decades, there has been a well-recognized shift towards decreasing sizes of primary testicular tumours [15,17,25,26,27]. This trend was also apparent when the present results were compared with the classical data reported from Toronto in 1966 with a median size of 5 cm in both seminoma and nonseminoma [28], and with the Dateca report from 1984 with mean sizes of 48 mm and 40 mm for seminomas and nonseminomas, respectively [29]. 

Of note, 13.6% of all tumours in the present study were within the subcentimeter range. During the last two decades, a plethora of studies reported on incidentally detected small testicular tumours; however, the relative frequency of such findings remained rather ill-defined [30]. A large study from the UK identified 81 (3%) subcentimeter masses among a total of 2681 patients with testicular neoplasms [16]. This figure is considerably lower than the 13.6% proportion observed in the present study. Selection bias may have contributed to the difference, as our institution represents a referral centre for testicular diseases. However, the difference may also partly relate to the well-documented time trend of decreasing tumour size, since the UK study recruited patients during 2003–2016, while our study included patients treated more recently (2012–2021). Differences regarding health system-related factors might have contributed to the difference as well. In a study from the U.S., 22 tumours (10.6%) out of 208 testicular neoplasms were shown to be smaller than 1 cm, which is quite close to our finding. However, that study reported lesion sizes measured sonographically, and the authors found that sonography may underestimate the true lesion size [31]. In a study from Turkey, a 13.9% proportion of subcentimeter tumours was reported, but, that study only evaluated tumours sized <3 cm [32]. Finally, a Swiss study reported a frequency of 3% of tumours sized <1 cm among 849 patients with GCT. However, that study enrolled only patients with germ cell tumours undergoing orchiectomy and excluded benign histologies and all cases with testis sparing surgery [33]. So, the true frequency of subcentimeter testicular tumours in that population was probably much higher. In light of the ever-growing clinical use of scrotal sonography in urologic practice and its ongoing technical refinements [8,9,34,35], the relative frequency of subcentimeter testicular tumours is probably in the range of 10% or more in contemporary series. This figure clearly underscores the clinical relevance of managing incidentally detected small testicular lesions.

### 4.2. Association of Primary Tumour Size with Histology (Assumption #1)

Benign tumours were shown to have a median size of only 10 mm, which is significantly smaller than that of GCTs (30 mm) and other malignant tumours (53 mm). More than 50% of all testicular tumours in the size category ≤10 mm consist of benign tumours, and we also noted a significant trend towards higher proportions of benign tumours with decreasing tumour size. Thus, small size of a testicular lesion appears to be a strong indicator of benign histology, and this result is in line with several previous reports [7,16,36,37,38,39,40,41,42]. The very high proportion of benign tumours among incidentally found small testicular masses had already been noted some decades ago [43], but this knowledge became clinically relevant only with the ever-growing number of small testicular neoplasms incidentally detected by improved ultrasonography technology. Accordingly, it was suggested to employ primary tumour size as a diagnostic tool for clinical assessment of incidentally detected small testicular masses [7,16]. A Turkish study of 252 patients including 35 cases with tumours ≤10 mm suggested a cutoff size of 15 mm to discriminate between benign and malignant tumours [32]. In the present study, the ROC analysis for prediction of malignancy by lesion size revealed a cutoff size of 16 mm to involve a sensitivity and specificity of 81.5% and 81%, respectively. Accordingly, the logistic regression curve revealed all tumours sized ≥ 39 mm to represent malignancy, while tumours ≤ 8 mm involved a 50% chance of being benign. Seven previous studies reporting results from ROC analyses are listed in Table 16 ([7,23,31,37,41,44,45]). Only one study reported on more than 100 patients [7]. Because of the divergent and mostly small sample sizes, the results are not consistent. Cutoff sizes ranged from 5 mm to 18.5 mm. Six studies reported sensitivities of >80%, while the others noted sensitivities of 55% or less, with AUC values ranging from 0.59 to 0.902.

Though all of these investigations principally support the potential value of tumour size in predicting testicular histology, there is currently no consensus about the threshold sizes for clinical decision-making [46]. 

Caution comes from a recent study from Switzerland reporting metastatic disease in 5 of 25 patients (20%) with small germ cell tumours [33]. The authors, therefore, challenged the view that testicular masses < 1 cm are of benign nature without fail. Undisputedly, malignant testicular tumours may occur in the subcentimeter size category. In the present study, the proportion was 49%, and this finding is consistent with several other reports on small testicular tumours [7,16,38,39,40,45,47]. However, the authors of the Swiss study probably overestimated the risk of metastasized malignancy in newly detected testicular subcentimeter masses because they included only GCTs in patients undergoing full orchiectomy in their analysis, without considering benign tumours and neoplasms managed with TSS. In aggregate, tumour size is probably a valuable tool for clinically assessing small testicular masses, although it certainly needs to be employed with caution. In practical terms, conservative surgery using intraoperative frozen section examination appears to be appropriate in small testicular masses, since malignancy can principally be encountered among the incidentally detected lesions. Surveillance could be a solution in the very small masses (<5 mm) [9,35,48,49,50]. 

### 4.3. The Impact of Tumour Size on Clinical Staging in Testicular Germ Cell Tumours (Assumption #2)

The present study documented a clear association of tumour size with clinical stage. In the size category ≤10 mm, 97% of cases had localized disease (CS1). The median tumour size increased with increasing clinical stages, with a size of only 30 mm in CS1 and 50 mm in CS3. Very similar results were reported from the classical Dateca series where 84% of seminomas were sized <2.5 cm and had CS1 disease, whereas in the largest size category (>8.5 cm), only 43% had CS1. Likewise, in nonseminomas sized <2.5 cm, 71% of cases had CS1, as opposed to 42% in the largest size category [29]. The Mannheim group reported mean tumour sizes of 37 mm and 42 mm in seminoma cases with CS1 and in those with > CS1, respectively [17]. On the other hand, the present data also show that even in the size category >30 mm, as many as 69% of cases had CS1, and conversely, only 10% of all CS1 cases had tumour sizes ≤10 mm. Thus, the usefulness of tumour size as an aid for clinical decision making is probably limited because even patients with large primary tumours may have localized disease (CS1). Accordingly, the Indiana series did not observe different frequencies of lymph node metastases in tumour size categories <2 cm, 2–5 cm, and >5 cm, respectively, among nonseminoma patients undergoing retroperitoneal lymph node dissection [51]. However, in seminoma, tumour size does appear to have a bearing on the metastatic potential of the tumour. In 2002, Warde et al. found a tumour size of >4 cm in testicular seminoma to involve a twofold risk of progression compared to smaller tumour sizes [52]. This significant association was confirmed by numerous clinical series and by two recent meta-analyses [11,53]. Accordingly, tumour size has been included as a prognostic factor for seminoma in all major guidelines [20,54,55]. It is unclear which particular biological mechanisms can induce metastatic spread (and thus higher clinical stages) in larger tumours. As tumour size is also associated with local pathological stage [19], it may be speculated that increasing tumour size may induce intratumoral processes promoting invasive tumour growth with metastatic spread. 

### 4.4. The Association of Tumour Size with Serum Tumour Marker Expression Rates in Germ Cell Tumours (Assumption #3)

Significantly higher expression rates of all serum tumour markers were observed in the tumour size category >10 mm than in smaller tumours (≤10 mm). As serum markers are secreted from both the primary tumour and its metastases, the biological impact of tumour size on marker expression is best evaluated in localized disease (CS1). Here again, marker expression rates were significantly higher in the larger tumour size category (>10 mm), with the exception of LDH. The non-significant difference between the tumour size categories regarding LDH expression rate may relate to the low specificity of this marker for germ cell tumours. 

In a more granular analysis evaluating marker expression rates in five tumour size categories, significant trends towards increasing marker expression rates with increasing tumour size were disclosed. This trend was found in both seminoma and nonseminoma, with the exceptions of AFP in seminoma (*p* = 0.9133) and M371 in nonseminoma (*p* = 0.0745). The two exceptions accord with expectations, since there was basically no AFP expression in seminoma, and hence no difference between tumour size categories regarding the expression of this marker. With respect to M371, the expression rate in nonseminoma was extremely high, with 66.7% even in the smallest size category; therefore, the difference relative to the larger size categories was comparatively small (100%). In light of the rather small sample size (*n* = 109, distributed over five size categories) the difference was no more significant, statistically. In all, these data clearly demonstrate the close association of serum-marker expressions with tumour bulk, and this result is consistent with the finding of higher serum levels of bHCG and AFP in patients with larger primary tumours than in those with smaller ones [12]. In another report, LDH expression rates were found to be higher in patients with larger tumours than in in those with small primaries [56]. In seminomas, bHCG serum levels were higher in patients with larger tumours than in those with small tumours [57]. In a small Spanish study, a significant correlation was found between tumour size and extent of elevations of bHCG and AFP [58]. With respect to clinical practice, the very low expression rates of bHCG and AFP (<10%) observed in the smallest tumour size category of germ cell tumours underscore the limited usefulness of the classical tumour markers for diagnosing subcentimeter testicular neoplasms. The novel marker M371 performed much better in that scenario, with expression rates of 66.7% and 39% in subcentimeter nonseminomas and seminomas, respectively. These results are widely in line with data reported from a multicentric study on 259 and 103 CS1 patients with seminoma and nonseminoma, respectively. In that study, M371 sensitivities of 56% and 98% were found for the diagnosis of subcentimeter seminomas and nonseminomas, respectively [14]. Support for the still-high sensitivity of M371 in small GCTs comes from a linear regression analysis reported from a Norwegian study on 131 patients with CS1 germ cell tumours. The authors found positive correlations between primary tumour size and M371 serum levels in both seminomas and nonseminomas, and quite a number of patients with subcentimeter primaries had measurable M371 levels. Notably, among the small tumours, a larger number of nonseminomas compared to seminomas had measurable M371 levels [59]. Overall, the sensitivity of M371 is better for nonseminoma than for seminoma regarding small testicular masses. However, in light of the almost negligible sensitivity of the classical markers in subcentimeter testicular neoplasms, the sensitivity of around 40% for M371 in small seminomas still represents significant progress, and probably, the M371 test can be a valuable tool for diagnosing small testicular neoplasms. 

### 4.5. No Association between Patient Age and Tumour Size (Assumption #4)

In the entire population of patients and also in the population of germ cell tumours, the largest median tumour sizes were observed in the oldest age category. Although the overall analysis revealed significant differences in median sizes among age categories, no clear trend of varying tumour sizes with age categories became apparent. Moreover, the youngest age category proved to present with the second largest tumour size, with only a 1.5 cm difference between the oldest and youngest category. These results are consistent with a previous investigation where no difference in tumour sizes had been found by comparing GCT patients <50 years with older ones [19]. Thus, the hypothesis of a trend towards larger tumours in the elderly, put forward by a UK-study, could not be confirmed by the present study [15].

### 4.6. Limitations of the Study

The retrospective study design could be a limitation to the present analysis because selection bias cannot entirely be ruled out. Small sample sizes in the histological subgroups BT and OM could have limited statistical power in spite of the overall large patient number in the study. Histological diagnoses relied on local pathological assessments only, with no central histopathological review. However, as both of the urologic departments participating in the present study had a clinical focus on testicular cancer, local pathologists clearly had suitable experience with testicular pathology. Misclassification of benign tumours cannot entirely be ruled out, since malignancy in some of these cases can only be documented by development of metastases during follow-up. However, as only one quarter of BTs had follow-up periods of less than 2 years, we believe that misclassification of BTs is probably only a minor problem. Missing information regarding therapeutic outcome might be another shortfall; however, the present investigation specifically aimed to analyse the biological role of tumour-size with regard to presenting features of testicular neoplasms. One possible strength of the study is the large number of patients with measurements of the novel tumour marker M371, featuring the utility of this test in clinical practice. Another asset could be the rather large number of patients with small testicular neoplasms, underscoring the utility of the factor of tumour size as a valuable tool in assessing such lesions.

## 5. Conclusions

The present study confirmed the significant biological role of tumour size in the clinical course of testicular neoplasms. In the diagnostic work-up of small testicular neoplasms, the factor of tumour size may be implemented clinically since more than 50% of cases with testicular new growths sized ≤10 mm had benign histologies. This knowledge may help in avoiding unnecessary orchiectomies and may open the door for a wider application of testis sparing surgery in these cases [60]. Increasing tumour sizes are associated with both increasing frequencies of metastatic disease (clinical stages > CS1) and increasing frequencies of elevation of serum tumour markers. Of note clinically, the novel marker M371 not only outperformed the classical markers in general, but was also expressed in around 40% of subcentimeter germ cell tumours. The present study did not look for the underlying molecular biologic processes that trigger the significant associations of tumour size with other clinical factors. However, it may be speculated that on the one hand, larger tumours involve a higher probability of the presence of marker-producing cells, and on the other hand, the larger number of proliferating cells involve a higher chance of developing cells with more aggressive features precipitating metastatic spread. The present study provides a multitude of data regarding descriptive features of testicular tumours that may frame future molecular biological investigations and may thus aid in understanding the underlying biological characteristics of this disease.

## Figures and Tables

**Figure 1 cancers-14-05447-f001:**
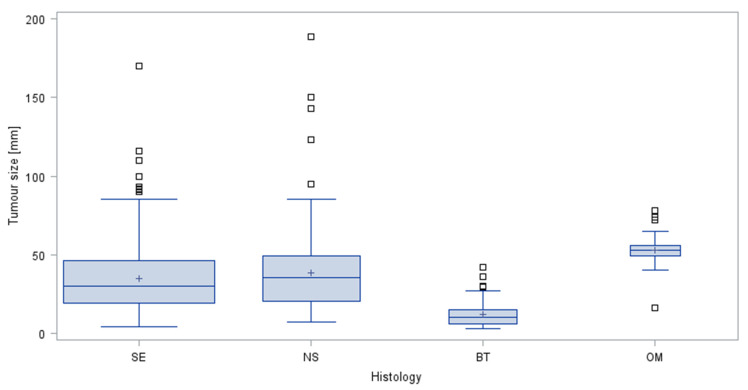
Tumour sizes observed in the four histologic groups. Box and whisker plots showing the distribution of tumour sizes stratified by histologic subtypes of testicular neoplasms. The boxes display the first quartile, median and third quartile. The whiskers are defined as the largest or lowest observed value that falls within 1.5 times the interquartile range measured from Q3 or Q1, respectively. Area of box relates to sample size. □ outliers; + denotes arithmetic mean; SE seminoma; NS nonseminoma; BT benign tumours; OM other malignancies.

**Figure 2 cancers-14-05447-f002:**
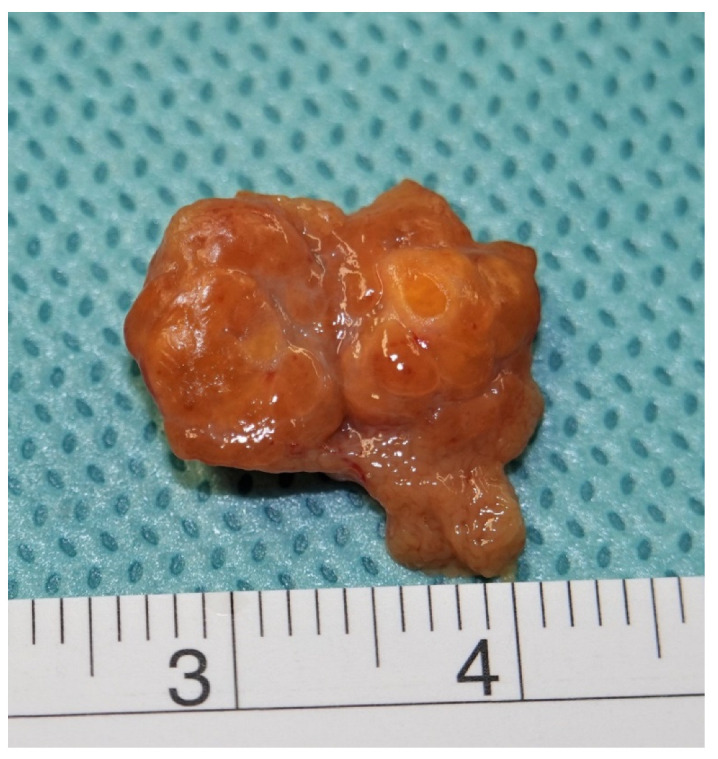
Typical subcentimeter testicular neoplasm. Surgical specimen of a 6 mm sized benign Leydig cell tumour excised by testis sparing surgery.

**Figure 3 cancers-14-05447-f003:**
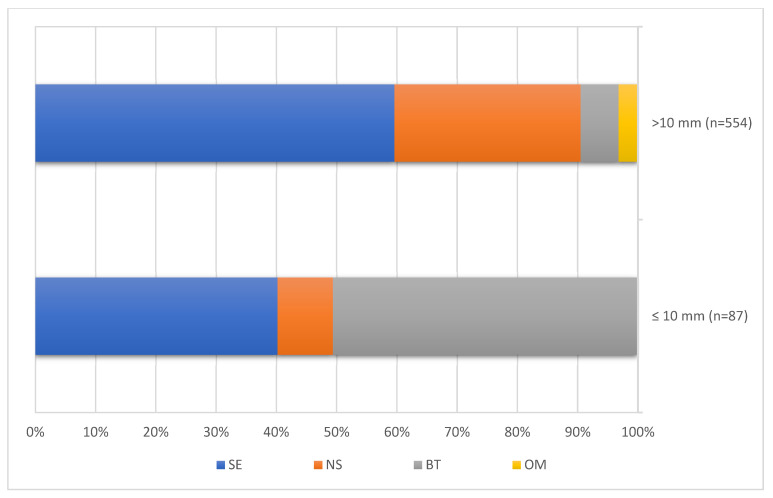
Proportions of histologic subgroups in testicular tumours sized >10 mm and ≤10 mm.

**Figure 4 cancers-14-05447-f004:**
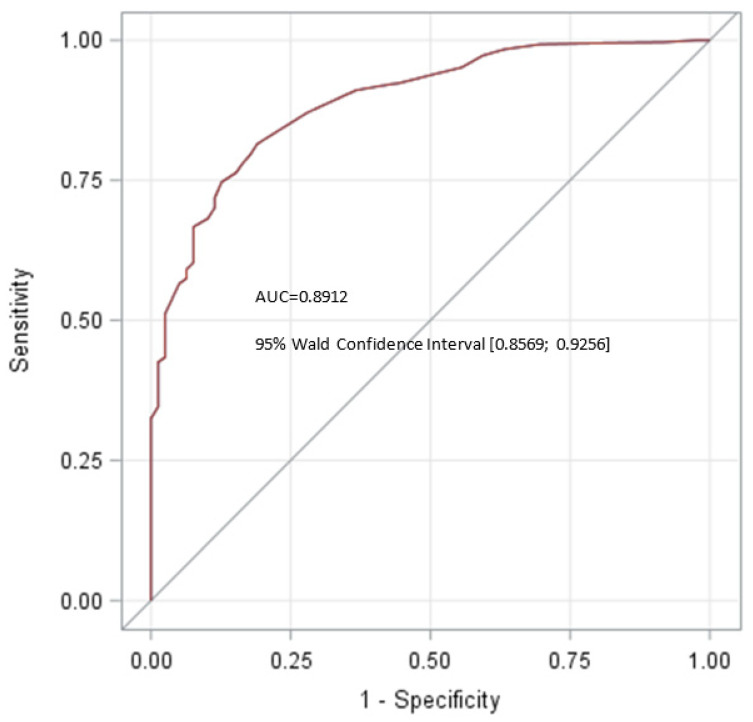
ROC analysis for predicting histology of a testicular neoplasm by its size.

**Figure 5 cancers-14-05447-f005:**
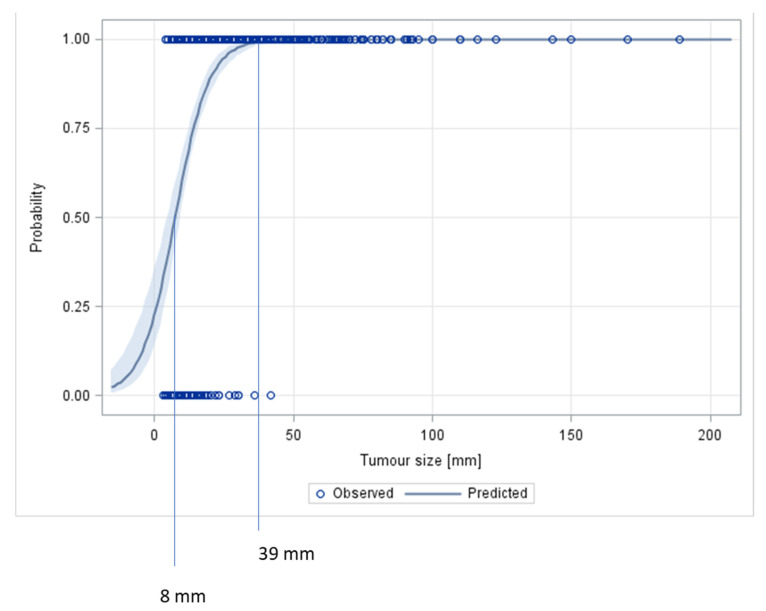
Probability curve for prediction of malignant histology of a testicular neoplasm. The logistic regression curve indicates the probability of a given tumour size to predict malignancy. Shadowed area represents 95% confidence intervals. Neoplasms with a size of ≤8 mm involve a 50% probability of malignancy, while tumour sizes of ≥25 mm, ≥33 mm, and ≥39 mm involve probabilities of 95%, 99%, and 100%, respectively.

**Figure 6 cancers-14-05447-f006:**
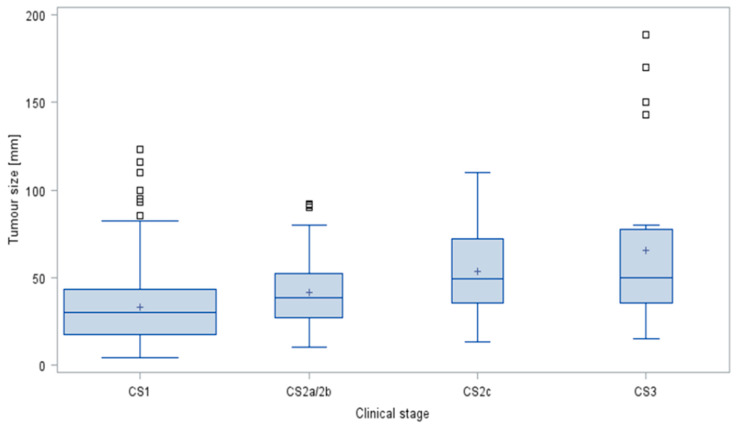
Tumour sizes in clinical stages in germ cell tumours. Box and whisker plots showing the distribution of tumour sizes stratified by clinical stages of testicular neoplasms. The boxes display the first quartile, median and third quartile. The whiskers are defined as the largest or lowest observed value that falls within 1.5 times the interquartile range measured from Q3 or Q1, respectively. Area of box denotes sample size. □ outliers; + denotes arithmetic mean; CS clinical stage.

**Figure 7 cancers-14-05447-f007:**
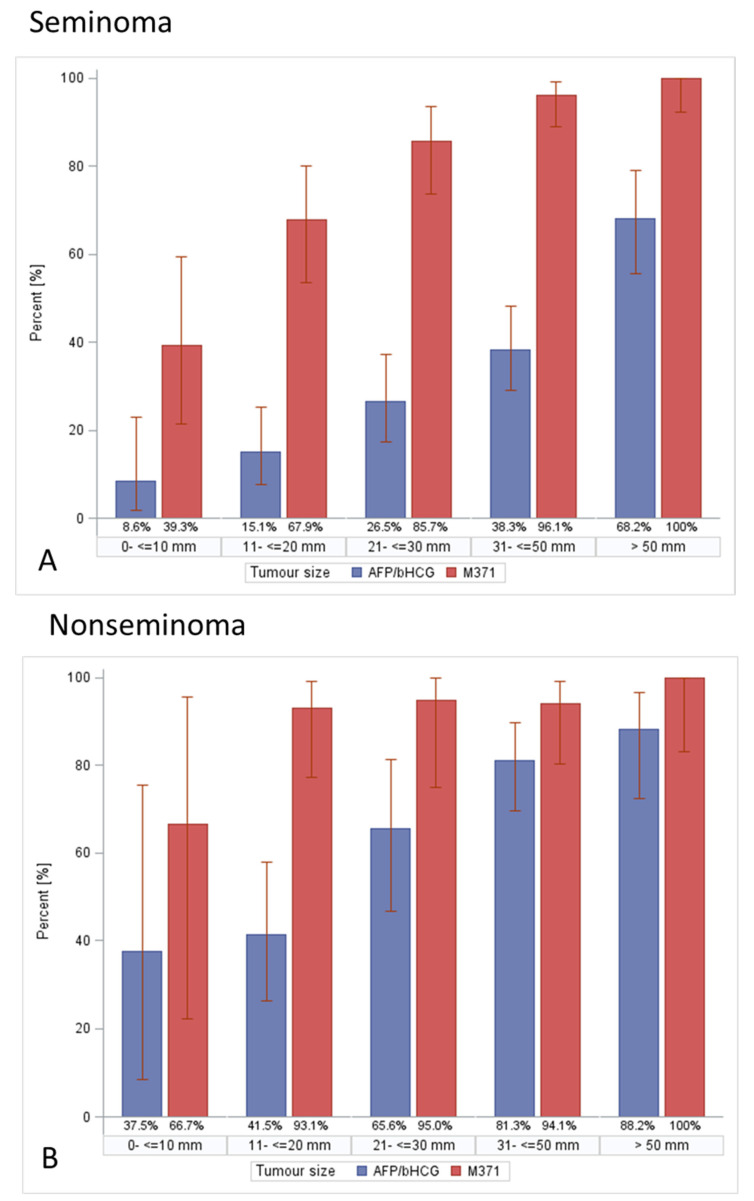
Expression rates of M371 and AFP and/or bHCG in germ cell tumours in relation to size of primary tumour in seminoma (**A**) and in nonseminoma (**B**). Blue columns denote expression rates of AFP and/or bHCG in five categories of tumour size, red columns indicate expression rates of M371. Overall, the expression rates of all tumour markers were higher in nonseminoma than in seminoma. All markers showed a significant trend towards lower expression rates with decreasing tumour size. M371 had higher expression rates than the other markers even in the smallest tumour size category. Error bars represent 95% CIs.

**Table 1 cancers-14-05447-t001:** Total patient population and frequency distribution of histologic subgroups and corresponding age distributions.

	n (%)	Age (Years)				
		Min	Q1	Median	Q3	Max
Total	641 (100%)	17	31	**38**	47	98
Seminoma (SE)	365 (56.94%)	17	33	**40**	48	78
Nonseminoma (NS)	179 (27.93%)	17	26	**31**	37	74
Benign tumours (BT)	79 (12.32%)	19	32	**41**	50	68
Other malignant tumours (OM)	18 (2.81%)	52	68	**72.5**	78	98

*n*: number of cases, Min: minimum, Q1: first quartile, Q3: third quartile, Max: maximum.

**Table 2 cancers-14-05447-t002:** Distribution of tumour sizes stratified by histologic subgroups.

	GCT (mm)	Seminomas (mm)	Nonseminomas (mm)	Benign Tumours (mm)	Other Malignant Tumours (mm)
Mean	35.9	34.8	38.2	11.8	53.0
Std Dev	23.2	22.2	25.7	7.8	14.3
Min	3	4	7	3	16
Q1	15	19	20	6	49
Median	30	30	35	10	53
Q3	46	46	49	15	56
Max	189	170	189	42	78
*n*	544	365	179	79	18

Min: minimum, Std Dev: standard deviation, Q1: first quartile, Q3: third quartile, Max: maximum, Overall comparison: *p* < 0.0001 (Kruskal–Wallis Test).

**Table 3 cancers-14-05447-t003:** Comparisons of median tumour sizes among particular histologic subgroups.

Histologic Groups	*p* *
SE vs. NS	0.1255
(SE + NS) vs. (BT + OM)	<0.0001
(SE + NS) vs. BT	<0.0001
(SE + NS) vs. OM	<0.0001
BT vs. OM	<0.0001

* Wilcoxon two-sample test.

**Table 4 cancers-14-05447-t004:** Frequencies of histologic subgroups in tumour size categories of ≤10 mm and >10 mm.

Size Categories	SE	NS	BT	OM	Total	*p* *
≤10 mm (*n* = 87)	35 (40.23%)	8 (9.20%)	44 (50.57%)	0 (0.00%)	100%	<0.0001
>10 mm (*n* = 554)	330 (59.57%)	171 (30.87%)	35 (6.32%)	18 (3.25%)	100%	

* Chi-squared test.

**Table 5 cancers-14-05447-t005:** Frequencies of histologic subgroups in four tumour size categories.

Size Categories	SE	NS	BT	OM	Total
≤10 mm (*n* = 87)	35 (40.23%)	8 (9.20%)	44 (50.57%)	0 (0.00%)	100%
11–20 (*n* = 141)	73 (51.77%)	41 (29.08%)	26 (18.44%)	1 (0.71%)	100%
21–30 (*n* = 123)	84 (68.29%)	32 (26.02%)	7 (5.69%)	0 (0.00%)	100%
>30 mm (*n* = 290)	173 (59.66%)	98 (33.79%)	2 (0.69%)	17 (5.86%)	100%

Overall comparison: *p* < 0.0001 (chi-squared test).

**Table 6 cancers-14-05447-t006:** Frequency distribution of malignant tumours and benign testicular tumours in four tumour size categories.

Size Categories	GCT + OM	BT	*p* *
≤10 (*n* = 87)	43 (49.43%)	44 (50.57%)	<0.0001
11–20 (*n* = 141)	115 (81.56%)	26 (18.44%)	
21–30 (*n* = 123)	116 (94.31%)	7 (5.69%)	
>30 mm (*n* = 290)	288 (99.31%)	2 (0.69%)	

* Cochran–Armitage trend test.

**Table 7 cancers-14-05447-t007:** Frequency distribution of clinical stages (CS) of germ cell tumours stratified by four categories of increasing tumour size.

	CS1 (*n*)	CS2a/2b (*n*)	CS2c (*n*)	CS3 (*n*)
≤10 mm	42 (97.67%)	1 (2.33%)	0 (0.00%)	0 (0.00%)
11–20 mm	98 (85.96%)	11 (9.65%)	3 (2.63%)	2 (1.75%)
21–30 mm	94 (81.03%)	19 (16.38%)	0 (0.00%)	3 (2.59%)
>30 mm	187 (69.00%)	53 (19.56%)	12 (4.43%)	19 (7.01%)

Jonckheere–Terpstra test to analyse increase in tumour sizes with increase in clinical stage: *p* < 0.0001.

**Table 8 cancers-14-05447-t008:** Frequencies of localized disease (CS1) and metastasized cases (>CS1) of germ cell tumours (SE, NS) in four categories of increasing tumour size.

	CS1 (*n*)	>CS1 (*n*)	*p* *
≤10 mm	42 (97.67%)	1 (2.33%)	<0.0001
11–20 mm	98 (85.96%)	16 (14.04%)	
21–30 mm	94 (81.03%)	22 (18.97%)	
>30 mm	187 (69.00%)	84 (31.00%)	

* Cochran-Armitage trend test.

**Table 9 cancers-14-05447-t009:** Distribution of tumour sizes stratified by clinical stages (CS) in germ cell tumours (SE, NS).

	CS1 (mm)	CS2a,b (mm)	CS2c (mm)	CS3 (mm)
Mean	32.6	41.2	53.2	65.5
Std Dev	20	19.5	30.1	48.3
Min	4	10	13	15
Q1	17	27	35	35.5
Median	30	38	49	50
Q3	43	52	72	77.5
Max	123	92	110	189
*n*	421	84	15	24

Min: minimum, Q1: first quartile, Q3: third quartile, Max: maximum, Std Dev: standard deviation.

**Table 10 cancers-14-05447-t010:** Serum tumour marker expression rates stratified by tumour size categories in the entire population of patients with testicular tumours (SE, NS, BT, OM).

	bHCG	AFP	LDH	AFP/bHCG	M371
	*n*/N	*n*/N	*n*/N	*n*/N	*n*/N
≤10 mm	6/87 (6.90%)	2/87 (2.30%)	3/87 (3.45%)	8/87 (9.20%)	16/75 (21.33%)
>10 mm	202/553 (36.53%)	116/553 (20.98%)	122/546 (22.34%)	241/553 (43.58%)	309/375 (82.40%)
*p*-value *	<0.0001	<0.0001	<0.0001	<0.0001	<0.0001

* Chi-squared test, AFP/bHCG elevation of AFP and/or bHCG; N: number of eligible patients.

**Table 11 cancers-14-05447-t011:** Serum tumour marker expression rates stratified by tumour size categories only in patients with germ cell tumours (SE, NS).

	bHCG	AFP	LDH	AFP/bHCG	M371
	*n*/N	*n*/N	*n*/N	*n*/N	*n*/N
≤10 mm	5/43 (11.63%)	1/43 (2.33%)	3/43 (6.98%)	6/43 (13.95%)	15/34 (44.12%)
>10 mm	299/500 (59.80%)	115/500 (23.00%)	119/493 (24.14%)	239/500 (47.80%)	302/335 (90.15%)
*p*-value *	0.0002	0.0015	0.0101	<0.0001	<0.0001

* Chi-squared test, AFP/bHCG elevation of AFP and/or bHCG. N: number of eligible patients.

**Table 12 cancers-14-05447-t012:** Serum tumour marker expression rates stratified by tumour size categories only in patients with germ cell tumours (SE, NS) and with clinical stage 1 (CS1).

	bHCG	AFP	LDH	AFP/ bHCG	M371
	*n*/N	*n*/N	*n*/N	*n*/N	*n*/N
≤10 mm	4/42 (9.52%)	1/42 (2.38%)	3/42 (7.14%)	5/42 (11.90%)	15/34 (44.12%)
>10 mm	132/378 (34.92%)	70/378 (18.52%)	63/375 (16.80%)	158/378 (41.80%)	240/273 (87.91%)
*p*-value *	0.0008	0.0081	0.1039	0.0002	<0.0001

* Chi-squared test, AFP/bHCG elevation of AFP and/or bHCG. N: number of eligible patients.

**Table 13 cancers-14-05447-t013:** Expression rates of serum tumour markers stratified by tumour size categories.

	bHCG	AFP	LDH	AFP/bHCG	M371
	*n*/N	*n*/N	*n*/N	*n*/N	*n*/N
**Total** **population**					
≤10 mm (*n* = 87)	6/87 (6.90%)	2/87 (2.30%)	3/87 (3.45%)	8/87 (9.20%)	16/75 (21.33%)
11–20 mm (*n* = 141)	23/141 (16.31%)	14 /141 (9.93%)	12/140 (8.57%)	30/141 (21.28%)	66/107 (61.68%)
21–30 mm (*n* = 123)	47/122 (30.33%)	21/122 (17.21%)	15/121 (12.39%)	43/122 (35.25%)	67/81 (82.72%)
31–50 mm (*n* = 179)	76/179 (42.46%)	52 /179 (29.05%)	45/176 (25.57%)	93/179 (51.96%)	107/114 (93.86%)
>50 mm (*n* = 111)	66/111 (59.46%)	29/111 (26.13%)	50/109 (45.87%)	75/111 (67.57%)	69/73 (94.52%)
*p*-value *	<0.0001	<0.0001	<0.0001	<0.0001	<0.0001
**Seminoma**					
≤10 mm	2/35 (5.71%)	1/35 (2.86%)	3/35 (8.57%)	3/35 (8.57%)	11/28 (39.29%)
11–20 mm (*n* = 73)	8/73 (10.96%)	3/73 (4.11%)	8/72 (11.11%)	11/73 (15.07%)	36/53 (67.92%)
21–30 mm (*n* = 84)	18/83 (21.69%)	5/83 (6.02%)	9/83 (10.84%)	22/83 (26.51%)	48/56 (85.71%)
31–50 mm (*n* = 107)	35/107 (32.71%)	8/107 (7.48%)	27/106 (25.47%)	41/107 (38.32%)	74/77 (96.10%)
>50 mm (*n* = 66)	43/66 (65.15%)	2/66 (3.03%)	35/66 (53.03%)	45/66 (68.18%)	46/46 (100%)
*p*-value *	<0.0001	0.9133	<0.0001	<0.0001	<0.0001
**Nonseminoma**					
≤10 mm (*n* = 8)	3/8/37.50%)	0/8 (0.00%)	0/8 (0.00%)	3/8 (37.50%)	4/6 (66.67%)
11–20 mm (*n* = 41)	14/41 (34.15%)	10/41 (24.39%)	3/41 (7.32%)	17/41 (41.46%)	27/29 (93.10%)
21–30 mm (*n* = 32)	19/32 (59.38%)	16/32 (50.00%)	6/31 (19.35%)	21/32 (65.63%)	19/20 (95.00%)
31–50 mm (*n* = 64)	41/64 (64.06%)	44/64 (68.75%)	17/62 (27.42%)	52/64 (81.25%)	32/34 (94.12%)
>50 mm (*n* = 34)	23/34 (67.65%)	27/34 (79.41%)	14/32 (43.75%)	30/34 (88.24%)	20/20 (100%)
*p*-value *	0.0022	<0.0001	<0.0001	<0.0001	0.0745

* Cochran–Armitage trend test, AFP/ bHCG elevation of AFP or bHCG or of both markers, N: number of patients eligible in subgroup.

**Table 14 cancers-14-05447-t014:** Distribution of tumour sizes stratified by age categories in all patients with testicular neoplasms (SE, NS, BT, OM).

Age Categories (years)	Tumour Size (mm)					*p* *
	Min	Q1	Median	Q3	Max	
≤30	3	18	**32**	50	189	0.0117
31–40	4	15	**28**	43	95	
41–50	4	14	**24**	40	110	
>50	3	15	**35**	53	170	

Min: minimum, Q1: first quartile, Q3: third quartile, Max: maximum, * Overall comparison between age groups: *p* < 0.0001 (Kruskal–Wallis test).

**Table 15 cancers-14-05447-t015:** Distribution of tumour sizes stratified by age categories in patients with germ-cell tumours, only (SE + SE).

Age Categories (years)	Tumour Size (mm)					*p* *
	Min	Q1	Median	Q3	Max	
≤30	6	24	**35**	50	189	0.0161
31–40	4	20	**30**	45	95	
41–50	4	15	**27**	42	110	
>50	7	19	**36.5**	53	170	

Min: minimum, Q1: first quartile, Q3: third quartile, Max: maximum, * Overall comparison between age groups: *p* < 0.0001 (Kruskal–Wallis test).

**Table 16 cancers-14-05447-t016:** Sensitivity and specificity of tumour size as diagnostic tools for assessing small testicular masses—Synopsis of studies using ROC analysis.

		Patients	Cutoff	AUC	95% CI	Sensitivity	Specificity
First Author [#]	Year	(*n*)	(mm)			(%)	(%)
Shilo [37]	2012	11	18.5	0.902		87	83
Paffenholz [23]	2018	28	14	0.896		83	89
Gentile [7]	2019	108	8.5	0.75	0.63; 0.86	81	58
Staudacher [44]	2020	60	13.5	0.726	0.623; 0.828	55	85
Schwen [31]	2021	22	10	0.60		31.8	88.7
Gobbo [41]	2022	56	10	0.59	0.43; 0.75	25	92.9
Del Real [45]	2022	22	18.3	0.753		83	74
Present study	2022	79	16	0.8912	0.8569; 0.9256	81.5	81.0

*n*: patients with benign histology included in study; (#) number in list of references.

## Data Availability

The datasets generated and analyzed in this study are available upon reasonable request.

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
