# Peer review of "Testicular Neoplasms: Primary Tumour Size Is Closely Interrelated with Histology, Clinical Staging, and Tumour Marker Expression Rates—A Comprehensive Statistical Analysis"

_cancers, 2022, doi:10.3390/cancers14215447_

Round 1

Reviewer 1 Report

The authors should be congratulated for the present study. I have only few minor remarks.

1. The authors considered that benign tumours are comprised of gonadal stromal tumours, benign epidermoid cysts and other rare tumour. It is important to note that these tumors are gonadal stromal tumours not invariably benign. Authors did not report any other potential pathologic malignant features of gonadal stromal tumours beyond tumor size and since follow-up was not reported, malignant forms of these tumors cannot be excluded with certainty. This should be stated in the limitation section.

 2. The table 1 header should be properly marked. Was there a 98 year patients in both seminoma and other malignant tumours group?

Author Response

Point #1 (limitation section: gonadal stromal tumours could be malignant)

The reviewer is right by saying that the malignant character of gonadal stromal tumours can only be ruled out by uneventful follow-up and thus misclassification is principally conceivable. However, the rate of misclassification of benign tumours in our series is probably low because firstly, characterization as benign tumours was not merely based on tumour-size but also on mitotic count, infiltrative growth pattern, increased mitotic activity, vascular invasion, necrotic areas, and nuclear atypia. Secondly, only one quarter of the benign tumours of our series had a follow-up of less than 2 years.  So, we agree that the issue raised by the reviewer may basically exist, however practically, the likelihood of confounding of results by this is likely very low.

We included the following sentence in the limitations section: “Misclassification of benign tumours cannot entirely be ruled out, since malignancy in some of these cases can only be documented by development of metastases during follow-up. However, as only one quarter of BTs had follow-up periods of less than 2 years, we believe that misclassification of BTs is probably only a minor problem.”

Point #2 (Header of table 1, line 159 of revised manuscript)

We agree that the header of this table could be misleading. We changed the head-line to: “Total patient population and frequency distribution of histologic subgroups  and corresponding age distributions”. To explain the 98 year old patient: There was in fact one 98-year old patient with malignant lymphoma include in our study.  This patient was listed as maximum value first in the total population (first line of table and again in the last line (other malignant tumours).

Reviewer 2 Report

The article is well written and well structured. The topic is interesting, given the scarcity of currently approved therapies for the cancer in question. Furthermore, it provides further confirmation of the role of primary tumor size with respect to histology, markers and clinical staging.

The article can be published in this form

Author Response

Thank you very much indeed for reviewing our manuscript “Testicular neoplasms: primary tumour size is closely interrelated with histology, clinical staging, and tumour marker expression rates - a comprehensive statistical analysis”. 

Reviewer 3 Report

This is a well written paper from Germany, that assesses the ability of testicular tumor mass size to predict histology, clinical staging and tumor makers analysis.   641 patients are analyzed, mostly ns and seminoma with only 79 benign tumors and 18 other malignancies.   The authors do a good job in the discussion conveying the weaknesses and strengths of their study and putting their findings in context with prior literature.   However, novelty is relatively low and their findings mostly confirm prior findings with little new information.   However the work is a good compilation of the literature and their confirmatory findings are with a relatively large data set that adds value.    In balance this is a good addition to the literature on the utility and limitations of tumor mass size as a clinical predictor. 

Minor, some typos and the legend to table 1 is miss-aligned.  

Author Response

thank you very much indeed for reviewing our manuscript “Testicular neoplasms: primary tumour size is closely interrelated with histology, clinical staging, and tumour marker expression rates - a comprehensive statistical analysis”. 

We have change the legend of table 1. We have described the legend of the table 1 more detailed.

Reviewer 4 Report

The authors present their experience with the management of testicular cancer over a period of almost a decade. Since testicular cancer is an orphan disease in urology, is very difficult to create solid guidelines or prospective evaluation of different testicular cancer histology.

 The manuscript is well written with good clinical hypotheses evaluated.

I agree with the authors that there is no trend for larger tumors in elderly and I’m glad that there is finally proof for that

No comments from my part.

Author Response

(The authors gave the same response as above.)
